# Fragment-Based Ligand-Protein Contact Statistics: Application to Docking Simulations

**DOI:** 10.3390/ijms20102499

**Published:** 2019-05-21

**Authors:** Gabriele Macari, Daniele Toti, Carlo Del Moro, Fabio Polticelli

**Affiliations:** 1Department of Sciences, Roma Tre University, 00146 Rome, Italy; gabriele.macari@uniroma3.it (G.M.); toti@dia.uniroma3.it (D.T.); delmoro.carlo@gmail.com (C.D.M.); 2National Institute of Nuclear Physics, Roma Tre University, 00146 Rome, Italy

**Keywords:** molecular docking, scoring functions, fragment-based contact statistics

## Abstract

In this work, the information contained in the contacts between fragments of small-molecule ligands and protein residues has been collected and its exploitability has been verified by using the scoring of docking simulations as a test case for bringing about a proof of concept. Contact statistics between small-molecule fragments and binding site residues were collected and analyzed using a dataset composed of 200,000+ binding sites and associated ligands, derived from the database of the LIBRA ligand binding site recognition software, as a starting point. The fragments were generated by applying the decomposition algorithm implemented in BRICS. A simple “potential” based on the contact frequencies was tested against the CASF-2013 benchmark; its performance was then evaluated through the rescoring of docking poses generated for the DUD-E dataset. The results obtained indicate that this approach, its simplicity notwithstanding, yields promising results that are comparable, and in some cases, superior, to those obtained with other, more complex scoring functions.

## 1. Introduction

Proteins exert their biological role by interacting with several different compounds, such as nucleic acids, ions, small molecules, and other proteins. Many biological functions of the proteins are regulated by the interaction between proteins and small molecules. The prediction and quantification of the strength of these interactions can be addressed by computational approaches, among which molecular docking simulations play a pivotal role [1].

Molecular docking simulations over the years have evolved, reflecting the changes in the protein-ligand interaction paradigm: initially, back when the commonly accepted model of interaction was the “lock and key” model postulated by Fisher, the interaction energies between a rigid protein and a rigid ligand were considered proportional to the geometric complementarity of their shapes [2]. When Koshland proposed the “induced fit” model, it was clear that the flexibility of both the ligand and receptor plays a pivotal role in the recognition event [3]. This research area has witnessed a bloom of countless docking tools using different search algorithms (incremental construction, genetics algorithms, shape complementarity, etc.) and scoring functions (knowledge-based, empiric potential, etc.); for additional insight on this topic, please refer to a more extensive review like [4] and references therein. The main focus of this work lies in the assessment of the information content of interaction patterns between small-molecule ligands’ fragments and protein residues. Several studies have already investigated the interaction patterns between proteins and other binding partners, although the rules that govern protein-ligand interactions have not yet been fully disclosed. In this regard, Thornton and coworkers identified the physico-chemical and geometric principles for four different types of protein-protein complexes [5]. Luscombe and colleagues observed non-covalent bond networks that apply across all protein-DNA interfaces [6]. Dudev and Lim summarized the principles that govern protein-metal ion interactions by identifying several rules with respect to the coordination mode, coordination number, metal selectivity, and coordination stereochemistry [7]. In addition, Soga and coworkers [8] identified protein amino acid preferences in ligand-binding pockets’ composition and proposed an index to rank the likelihood of a cavity being a ligand binding site. Recently, Chen Cao and colleagues [9] enriched this metric by including solvent exposure and dihedral angle preferences, showing improvement in binding site prediction. Following these studies, the purpose and originality of this work lies in the identification of common patterns of interaction between ligands’ fragments and protein residues through a systematic study of a large collection of available, experimentally-resolved protein-ligand complexes. In particular, the aim of this work was to study the interactions between the chemical components of the ligands and the protein microenvironment surrounding them. In order to do this, the semi-exhaustive collection of binding sites employed by the LIBRA binding site recognition software [10] was filtered, clustered, and fragmented. Furthermore, in order to assess the information value contained in the contact statistics derived, the scoring of docking simulation was taken as a test case for a proof of concept. To this aim, a simple scoring function based on these contact frequencies was tested against the 2013 Comparative Assessment of Scoring Functions (CASF) benchmark [11]. Furthermore, the performance of the method in the reranking of the docking poses obtained with AutoDock Vina on the Directory of Useful Decoys-Enhanced (DUD-E) database [12,13] was also evaluated.

## 2. Results

The results obtained in this work are presented in separate sections in the following paragraphs. In the first section, the results of ligands’ fragmentation using two different methods (i.e., Breaking of Retrosynthetically Interesting Chemical Substructures (BRICS) [14] and REtrosynthetic Combinatorial Analysis Procedure (RECAPS) [15]) are presented, highlighting the reasons that led to choosing the BRICS fragmentation strategy. In the following section, the construction of different fragment-residue contact statistics datasets is described. Next, the performance of the various datasets, built in the previous phase, tested against the CASF-2013 benchmark, is reported. Lastly, in light of the results obtained on the CASF-2013 benchmark, a test of the performance of the contact statistics method in the re-ranking of the docking poses obtained with AutoDock Vina on the DUD-E database is reported and two additional virtual screening trials are analyzed and described.

### 2.1. Comparison of Fragmentation Methods

In order to choose the most suitable method for the purposes of this work, the RECAP and BRICS fragmentation methods were compared. The comparison involved the fragmentation of a non-redundant test set of ligands contained in the LIBRA database and derived from the Protein Data Bank [16] (see Methods section). Both tools are implemented in RDKIT (ver. 2018.09.1) [17]. The results of this analysis, reported in Table 1, indicate that BRICS achieves a better performance with a failure rate of 22.37% with respect to 50.84% achieved by RECAP. The distribution of fragments produced by the two methods (Figure 1) indicates that BRICS generates smaller fragments, with more than 80% of the fragments being smaller than 15 atoms.

Looking in detail at the fragments generated by the two methods (Figure 2), one can observe how RECAP links together smaller fragments of the same ligand to generate larger substructures. This behavior, though able to retain more “drug-like” features and to sample more chemical space, introduces, for the purpose of this study, an unwanted element of redundancy. In fact, smaller fragments are expected to provide a higher residue-fragment contact specificity. In light of these results, BRICS was chosen for the fragmentation of the ligands dataset. An additional advantage of the latter approach is that the bonds cleaved by BRICS are tagged with an isotopic label which identifies the specific chemical environment of each fragment in the context of the whole ligand. This gives the possibility to use the isotopic label for fragment rejoining and drug design applications, due to the retrosynthetic nature of the algorithm employed by BRICS.

### 2.2. Initial Dataset Building and Statistics

Following ligand fragmentation, residue-fragment contact frequencies were calculated for eight different initial datasets. Two datasets, CS_40 and CS_50, were obtained by applying proteins’ sequence identity threshold to filter out similar proteins and avoid a bias in the dataset (CS stands for “Contact Statistics” and the two digits refer to the proteins’ sequence identity threshold in %). In this case, only the ligand corresponding to the centroid of each protein cluster has been included in the dataset. The other six datasets feature different combinations of the parameters related to the proteins’ identity threshold and the ligand similarity. The datasets have been named CS_5003, CS_5005, CS_5007 and CS_4003, CS_4005, and CS_4007. Here, the first two digits refer to the proteins’ sequence identity threshold, while the last two digits refer to the Jaccard distance threshold employed to include all sufficiently dissimilar ligands. For instance, CS_5003 includes all protein ligand complexes for which the maximum sequence identity is 50% and, for each protein cluster, all ligands displaying a Jaccard distance higher than 0.3. The number of unique ligands and proteins and the number of fragments for each dataset are reported in Table 2. 

In the table, the last column indicates how many fragments are featured more than 20 times in the dataset. This value is important because the contact frequencies were only calculated for these fragments to ensure the statistical significance of each residue-fragment frequency. Analysis of the amino acids contained in the initial dataset revealed an amino acid composition in agreement with the results obtained by Soga and co-workers [8]. In particular, binding sites display a significant deviation in amino acid composition with respect to that of the proteins they are derived from and that of all the known proteins contained in UniProt [18] (Figure 3).

In order to observe how much the calculated contact frequencies deviate from a random distribution, for each fragment in the database, the contact frequencies with each of the twenty aminoacids were calculated and sorted in descending order. Then, these frequency arrays were summed up and averaged (Figure 4; see Methods for details). If the observed frequency distribution of the contacts was random, the averaged frequencies would be expected to be close to each other and around 5%. The results indicate, instead, that there is a significant deviation from this value. This suggests that fragments display a clear tendency to be in contact with only a limited set of residue types, hinting at a relevant information value contained in the fragment-residue contact frequency.

### 2.3. Results of the Tests on the CASF-2013 Dataset

To check whether fragment-residue contact frequencies can be used as a predictive index, they have been tested against the CASF-2013 benchmark. In order to keep the approach as simple as possible, for each predicted protein-ligand complex, fragment-residue contacts were collected and the score was simply calculated as a sum of the contact frequencies derived from the various contact statistics (CS) datasets (see Methods, Equation (1)). Here, the main interest was to verify the presence of exploitable common patterns in fragment-residue contacts and assess the limits of this approach, rather than focus on its sheer performance. To avoid any bias, the 195 proteins present in the CASF-2013 benchmark have been removed from the starting datasets used to calculate the frequencies. For consistency, the data obtained have been analyzed using the same scripts provided by the CASF’s authors. The results of the method using the different contact frequency datasets are reported below.

#### 2.3.1. Docking Power

Docking power assesses the scoring function’s ability to correctly identify, for each target, the native binding pose of a ligand among a set of decoy poses. A pose is deemed native if its root-mean-square deviation (RMSD) is lower than 2 Å with respect to the co-crystallized ligand pose. The pool of poses screened includes the co-crystallized ligand together with other poses predicted in silico. The results of the docking power test, expressed by the number of predictions in which the native pose is recognized as being the first, second, or third ranking pose, are shown in Figure 5.

The method proposed in this work (named, from now on, Contact Statistics (CS)) was able to correctly recognize the native pose as the first ranking pose in 30.3% of the cases, using the dataset built with a protein identity threshold of 50% and a Jaccard distance of 0.5 (i.e., dataset CS_5005). As can be observed in Table 3, CS_5005 performs better than the other datasets with respect to the second-best ranking pose (with a success rate of 40.5%) and the third-best ranking pose (with a success rate of 52.3%). With respect to other scoring functions, all CS implementations achieve better results than DS@Sybyl [19] and the simple dSAS (delta Solvent Accessible Surface) [11] score, with the latter representing the variation of the solvent accessible surface upon binding.

#### 2.3.2. Ranking Power

The ranking power test assesses the ability of a scoring function to correctly rank the ligands of a target protein by their binding affinity, given the native poses of these ligands. In the CASF-2013 benchmark, for each target, there are three ligands with a different binding affinity. It is possible to identify the best binder, the poorest binder, and the intermediate binder. Performances are divided into a “high-level” success rate and “low-level” success rate. The former measure counts the number of targets for which the three complexes are correctly ranked (best > median > poorest). The latter requires that only the ligand with the best affinity is correctly ranked (best > median and best > poorest). The results are graphically represented in Figure 6. CS_5003 and CS_4003, with a success rate of 23.1% in the high-level prediction and a success rate of 49.2% and 50.8% in the low-level prediction, respectively, are the best performing implementations of the CS method. Compared with other scoring functions, their performance puts them ahead of GlideScore-XP [20] in the low-level prediction. However, they achieve a poorer performance in the high-level prediction. The different performance between high-level prediction and low-level prediction indicates that this approach has more issues at handling medium-affinity and low-affinity complexes. Nevertheless, these results suggest that the most profitable contacts in terms of binding affinity are well-characterized and conserved.

#### 2.3.3. Scoring Power

The scoring power measures the ability of a scoring function to produce results linearly correlated with binding constants experimentally determined. The results related to this metric are shown in Table 4. The idea was to explore the possibility of correlating scoring results with experimental binding affinity, even though the simplicity of the proposed scoring scheme, being a simple sum of contact frequencies, can hardly reflect such complex metrics. The CS’s simple scoring scheme succeeds in capturing a correlation, although a weak one, between the score produced and the binding constant experimentally determined. In particular, the CS_5003 dataset achieves a Pearson correlation coefficient of 0.252 and a standard deviation of 2.26 log K_a_, showing a stronger correlation compared to London-dG@MOE and PMF@SYBYL. Among the different CS implementations, it is possible to observe the detrimental effect on the performance of lowering the Jaccard distance threshold of the ligands (in building the initial dataset). This observation reflects the results of the ranking power test, in which the best performers are the two implementations of CS with the tightest Jaccard distance threshold. This result can be explained by the inclusion of low-similarity ligands in each protein cluster that tends to lower the specificity of the contacts captured by the fragmentation approach. Likewise, the poor performance of CS_40 and CS_50 is due to the low number of fragments contained in these datasets, which translates into a low coverage. It is worth noting that of the 195 structures used to build the CASF-2013 benchmark, the method proposed in this work is only applicable on 146 complexes. This is due to the fact that in 49 cases, BRICS fails to correctly decompose the input ligand. By analyzing these compounds, it is apparent that these ligands are made up of rings fused together. This particular geometry is recognized as a unique fragment, due to how BRICS handles ring substructures.

#### 2.3.4. Screening Power

The screening power evaluates the ability of a scoring function to distinguish, for each target, true binders from random ligands, emulating a virtual screening experiment. In this test, the decoys for each target are represented by the other targets’ true binders, for a total of 12.675 ligand-protein pairs. For each docking-pair, there are up to 50 poses generated through GOLD [21], Surflex [22], and MOE [23]. The test offers a true binder/decoy ratio of 1:64. The results are expressed in terms of the enrichment factor, which represents the ratio between true binders observed in the top n% of scored poses and the number of true binders expected in the same portion of the sample (see Methods, Equation (3)). A higher enrichment factor is correlated with a higher probability of finding true binders among the top-ranked elements in virtual screening experiments. The results reported in Figure 7 show an interesting scenario. 

First, the results of all the CS implementations were unsatisfying, with the method presented here only performing better than dSAS. By looking at these results, a re-evaluation of the scoring approach presented here was prompted. In virtual screening experiments, different compounds are tested against the same receptor. Among these ligands, there are, obviously, both true and false binders. In this test, the length of the ligands (and hence the number of fragments) can be very different; thus, comparing them based on an additive scoring scheme can be misleading. For these reasons, an average of the contact frequencies observed for each binder’s fragments was deemed to be a more suitable metric to tackle this task. The re-evaluated CS scoring functions perform much better than the previous ones. In particular, CS_05 avg scores an enrichment factor in the top 1% of 4.61, a striking result which puts the method proposed here ahead of several other methods. Details of the scoring are presented in Appendix A (Appendix A).

### 2.4. DUD-E Results

After the large-scale tests carried out on the CASF-2013 benchmark, the validity of the method was tested by rescoring the results of molecular docking simulations performed on the DUD-E dataset using AutoDock Vina as a docking engine. Following the results obtained in the screening power test, the “averaged frequencies” version of the CS method was used in the DUD-E tests. All the different DUD-E subsets divided into protein families have been used as the test set. In the first test, re-docking of the co-crystallized ligand has been performed, and the generated poses have then been rescored by applying the averaged contact frequency. Then, for the top scoring poses for one method, i.e., CS, and the other, i.e., AutoDock Vina, the RMSD with respect to the co-crystallized ligands has been calculated. If the RMSD was lower than 2.0 Å, the pose was considered native; otherwise, the difference with the RMSD obtained for the top scoring pose was calculated. In the second test, two structures on which the two methods had a bad performance were selected and used for a virtual screening simulation. In this simulation, 20 active compounds and 400 decoy compounds were selected for each target. The scores obtained from the virtual screening were ranked and the enrichment factor was evaluated using the same thresholds employed in the CASF-2013 benchmark (i.e., 1%, 5%, 10%). The results are shown in Table 5. Regarding the RMSD differences between the top scoring pose of AutoDock Vina and the ones obtained by the CS method, the best performing starting dataset is CS_4005 (Figure 8). In fact, with this dataset, in nearly 52% of the cases, CS performs better than or equal to AutoDock Vina. Furthermore, in the 42 cases in which AutoDock Vina performs better than the CS method, the ΔRMSD for 10 of them is under 1Å. On the other hand, it must be noted that in most of the cases in which AutoDock Vina performs better, the ΔRMSD for the CS method is consistently higher than that observed in the opposite situation.

The performance of the other datasets is similar, with the worst performance being achieved by CS_4007 (Appendix A, Appendix A). One of the highest ΔRMSD scores between AutoDock Vina and the approach proposed in this work was obtained on the proteins mmp13 (CS performing better than AutoDock Vina; ΔRMSD = 13.9 Å) and urok (AutoDock Vina performing better than CS; ΔRMSD = 25.1 Å). In the first, the top scoring pose identified by the approach proposed here is closer to the co-crystallized ligand (Figure 9A), while AutoDock Vina places its top scoring pose far away from the binding pocket. On the other hand, in urok, the situation is reversed, with the prediction of AutoDock Vina being more accurate (Figure 9B).

Following the opposing results obtained on these two targets, the following test involved mmp13 and urok receptors, which were chosen as targets for a virtual screening trial (see Methods section for details). The results of the virtual screening are reported in Table 5. In this test, the enrichment factor calculated by Equation (3) (see Methods section) was used to assess the success rate of a scoring function. On mmp13, the CS method performance was striking, with an enrichment factor in the top 1% of 15 for almost all the fragment contact datasets, while AutoDock Vina was unable to identify any true binder. Good performances were also achieved on urok, with an enrichment factor of 15 obtained with both CS_5003 and CS_4003. The differences in the compositions of the different datasets are reflected in the enrichment factor obtained. In fact, in both mmp13 and urok, the smallest datasets, CS_50 and CS_40, performed worse than the others (Table 5). 

## 3. Discussion

The core contribution of this work was proving that significant and exploitable information content is associated with recurrent patterns of specific contacts between ligand fragments and binding site residues. Soga [8] and Cao [9] proposed the use of preferences displayed by amino acids to form a binding pocket as a descriptor to recognize ligand binding pockets. Inspired by these results, a simple approach was applied in this work to assess the existence of common residue-fragment interaction patterns in different protein-ligand complexes. The aim was to test the hypothesis that even a “trivial” approach, in the presence of a clear signal, can evidence the information content of the fragment-residue contacts. The results of the analysis performed on the initial datasets provide evidence of the tendency of ligands’ fragments to be in contact with only a limited number of residues (Figure 4). The amino acid composition of the binding sites identified by the fragment-residue contacts of the initial datasets is similar to the one obtained by Soga and coworkers. Indeed, it is possible to observe an over-representation of a limited number of residues, namely ASP, TYR, SER, ARG, and THR, together with the under-representation of CYS, MET, and PRO. While there are several similarities, there are also discrepancies, as in the cases of TRP and PHE residues. These discrepancies can be explained in light of the methodological differences between the two approaches. In the work by Soga and coworkers, all the residues surrounding a ligand within 4.5 Å are considered part of the binding sites; this work, instead, only takes into account the residue closest to the ligand fragment within 4.5 Å. Furthermore, the datasets produced in this work are made up of a much higher number of complexes, which are characterized by a significantly different chemical nature, with respect to the 41 protein-drug-like compound complexes in Soga’s dataset. As a consequence, the datasets described in this work can be considered a better representation of the binding sites landscape of the Protein Data Bank, given the higher chemical variety of ligands included. To assess whether the interaction patterns considered can be a valuable descriptor able to identify potential binders, docking tests have been carried out. The CASF-2013 benchmark was chosen for this purpose because it enabled experimentation of the proposed approach with respect to four different metrics: docking power, scoring power, screening power, and ranking power, and in comparison to other well-established scoring functions. Furthermore, this test decouples scoring functions from the search algorithm, allowing the test to focus only on the performance of the proposed descriptor. For this task, a naïve scoring function based on the frequencies of the residue-fragment contact identified has been developed. Even though a more sophisticated scoring function can obviously be developed, in this phase, this work was focused on the descriptors exactly as they are, in order to analyze the information value they provide without introducing additional variables.

Scoring power and ranking power tests both evaluate the ranking ability of the scoring functions. The first is focused on the ability to generate scores in linear correlation with the experimentally-determined binding affinity, whereas the second focuses on correctly ranking a set of three different affinity binders or at least on assigning the highest score to the strongest binder of the group. The two tests are interrelated and the proposed approach performs quite similarly in both. In CS_4007 and CS_5007, it is possible to observe a performance reduction by increasing the ligand’s Jaccard distance threshold. A higher Jaccard distance threshold corresponds to the formation of a lower number of clusters and leads to the inclusion of smaller numbers of complexes with a higher chemical variance in the final dataset. This variance leads to a partial loss of the information contained in the fragments’ contacts, resulting in performance degradation. This observation seems to confirm that the most profitable contacts in terms of binding affinity are well-characterized and conserved among similar ligands and similar fragments. Conversely, the docking power performance seems to reward a more inclusive clustering approach in the building of the dataset. Indeed, the best performing (even though just slightly) implementations of CS are those using a distance threshold of 0.7 or, at most, 0.5.

Screening power prompts some intriguing discussion. Indeed, the incremental scoring scheme fails to distinguish between true and false binders. One possible explanation, partially confirmed by the observation of the top scoring compounds, is that the length of the ligands can influence this kind of test. Indeed, when comparing them in a virtual screening scenario, an additive scoring scheme can be misleading. For these reasons, a scoring scheme based on the average of the contact frequencies calculated for each binder’s fragments, rather than the sum of them, could be a more suitable metric to tackle this task.

The tests carried out against AutoDock Vina in the DUD-E dataset show an overall good performance. In this case, the search algorithm employed is the one of AutoDock Vina, representing a crucial difference with respect to previous tests, in which the docking poses are already pre-calculated by Gold, Surflex, and MOE. In the blind rescoring test, the method is able to successfully detect a pose within 2.0 Å of RMSD from the co-crystallized ligand, even when the top scoring pose identified by AutoDock Vina is located far away from the binding region. This is another confirmation of the bias in amino acid composition which characterizes these regions. The results of the virtual screening test against AutoDock Vina show a good performance in both of the two screening targets selected. In particular, the best performance has been achieved, as expected, on mmp13, one of the proteins on which the CS method shows the best results in identifying a native-like pose. In fact, on mmp13, the enrichment factor in 1%, shared among all the starting datasets, is about 15 (with the exception of CS_5007, which achieves an enrichment factor of 10). The notable exceptions are CS_40 and CS_50, both yielding an enrichment factor in 1% of 0. The same happens on the other target, urok, on which the more numerous datasets obtain an enrichment factor of 15/10 in the top 1%, while CS_40 and CS_50 achieve an enrichment factor of only 5. Analyzing their fragment composition and that of the active compounds in detail, it is evident that some fragments are missing in both datasets, while the same fragments are present in the other datasets. The sample dimension of the datasets and the coverage of the ligands’ chemical space of the fragments in them are two crucial aspects of the method proposed here. The limited pool of fragments represents a weakness and a limitation of this approach. However, with the number of crystallized structures increasing every year, the applicability and coverage of statistical methods, such as that presented here, are set to increase in the same way. It must be kept in mind that this work was not meant to present a novel scoring function; instead, it was focused on trying to capture the information stored in the contacts collected in the PDB and verify the exploitability of this information, taking the scoring of docking simulations as a test case for a proof of concept. The results obtained both in the CASF-2013 benchmark and the DUD-E dataset indicate that the information underlying the fragment-residue contacts is valuable and can be exploited not only in molecular docking simulations. A better understanding of the interaction patterns of these moieties can lead to improved ligand binding prediction, protein function recognition, and drug design tools.

## 4. Materials and Methods 

In this work, contacts between fragments and binding site’s residues were analyzed using a starting dataset derived from the ligand binding sites database of LIBRA [24]. The fragments were generated by applying BRICS [14] decomposition as implemented in the Python chemoinformatic library RDKIT (ver. 2018.09.1) [17]. Dataset analysis and contact identification scripts were all written in Python and are available for download at the following URL: http://computationalbiology.it/software/macari_et_al_python_scripts.zip. The prediction performance of the contact frequency method was tested against the CASF-2013 benchmark [11] and its performance was evaluated through the rescoring of docking poses generated on the DUD-E dataset [13].

### 4.1. Fragmentation Methods

Fragmentation methods can be broadly divided into two categories based on the set of rules they apply for breaking ligand bonds: retrosynthetic methods and structure-based methods. The former methods break synthetically accessible bonds, while the latter identify target substructures, breaking the bonds between them. In order to ensure the most “objective” approach possible and the chemical significance of the generated fragments, a retrosynthetic approach was chosen. Two of the best regarded approaches of this kind are RECAP (REtrosynthetic Combinatorial Analysis Procedure) [15] and BRICS (Breaking of Retrosynthetically Interesting Chemical Substructures) [14]. RECAP is a molecule decomposition method based on the cleavage of 11 common molecular bonds. Susceptible bonds are cleaved in a single pass, avoiding the formation of intermediate structures. However, a terminal fragment is left uncleaved if it contains only small functional groups, such as methyl, ethyl, or hydrogen [15].

BRICS fragmentation is based on a set of 16 synthetic rules and applies all possible retrosynthetic cuts simultaneously, avoiding the generation of overlapping fragments. In addition to retrosynthetically splitting relevant bonds, it includes substructure filters in the ligand decomposition to avoid the generation of unwanted chemical motifs (e.g., hydrogen and halogen atoms, isopropyl groups, etc.) [14]. Both methods leave ring substructures uncut. To choose the most suitable method for the purposes of this work, a comparison, involving the fragmentation of a non-redundant ligands test set derived from PDB, was carried out. In this work, the two methods implemented in RDKIT were used.

### 4.2. Building of the Datasets

The fragment-residue contact datasets were derived from LIBRA’s binding site database (whose last update dates back to 2017) [24]. LIBRA’s binding site is a semi-exhaustive binding site database derived from PDB [16] containing 209,431 binding sites. Each entry, identified by a unique code (defined as XXXX_YYY_Z where XXXX is the PDB ID, YYY the ligand ID and Z the chain letter), contains a ligand and its surrounding amino acids within 5 Å. A ligand’s neighbor residues are divided into binding residues and environments. A preprocessing step involving the removal of incorrectly-formed entries was carried out, since a number of entries were either empty or only contained heteroatoms. In addition, to remove data redundancy, and ensure data accuracy and the biological relevance of the contacts identified, three different filtering procedures were carried out. In order to better underline the contact between a ligand’s fragment and the binding site’s amino acids, only proteins with high-quality crystal structures were included. This was achieved by filtering out complexes solved with a resolution higher than 2.5 Å. The resolution data of the protein structures were derived from the PDB data summary section in the form of a table containing both PDB ID codes and the corresponding resolution values. Complexes whose structures were determined by NMR were not included in the dataset. To ensure the biological significance of the identified contacts, non-biologically relevant compounds such as crystallization additives were removed from the datasets. The filtering procedure was performed according to a blacklist of compounds derived from BioLip [25]. In addition, entries containing carbohydrates, oligopeptides, ions, and nucleic acids, not relevant for this study, were manually removed. Lastly, to avoid bias due to over-represented proteins in the dataset, a sequence clustering procedure was performed using CD-HIT [26] and two different identity thresholds: 50% and 40%. Two complexes were kept even when their sequence identity was above the threshold if they sufficiently bound different ligands. In order to do this, among each protein cluster, the ligands were grouped based on their similarity. The similarity was measured through the Jaccard distance calculated on their circular Morgan fingerprints [27] and stored on a distance matrix. Based on this matrix, a Taylor-Butina clustering algorithm [28] was applied and the protein complex containing the centroid of each ligand cluster was added to the final database. For the ligands clustering phase, three different Jaccard distance thresholds were employed: 0.3, 0.5, and 0.7. When an entry contained multiple equivalent binding sites for the same ligand, only one binding site was considered, adding the information concerning additional residues present in the other binding sites.

### 4.3. Fragment-Residue Contact Identification

The contact identification procedure involved the fragmentation of all the dataset’s ligands. For each binding site, the ligand was fragmented using the RDKIT implementation of BRICS [14,17]. The data manipulated through a Python script were stored in a dictionary containing the code of the binding site as the key and the set of fragments generated from the fragmentation as the values. Then, each binding site residue was depicted as an ensemble of tagged points (depicting non-hydrogen atoms, with the tag representing the origin residue). A KD-Tree algorithm, implemented in the Scipy library [29], was employed for the spatial searching phase. Starting from the coordinates of each fragment, a residue was considered a neighbor of the fragment if there was at least one tagged point belonging to it, within a distance of 4.5 Å from a fragment’s atom. Only the closest residue of the protein was deemed in “contact” with the fragment. Contacts were registered in 20-bit arrays, with each bit representing a different amino-acid. For each fragment, the contact array was stored in a dedicated dictionary using the fragment’s SMILES as the key [30]. The contact frequencies were obtained by summing up contact arrays sharing the same SMILES and dividing the result by the number of occurrences of that fragment. The graph reported in Figure 4 has been obtained as follows: for all the fragments in the database, the frequency of contacts with each of the twenty amino acids was calculated and sorted in descending order. Then, these 1×20 frequency arrays were summed up and averaged, resulting in a frequency distribution to highlight how much this distribution deviates from a random one.

### 4.4. CASF-2013

The Comparative Assessment of Scoring Function (CASF) 2013 benchmark [11] is a test set made to evaluate docking scoring functions with respect to docking power, ranking power, scoring power, and screening power. It is derived from PDBBind [31] and includes 65 target proteins complexed with three ligands characterized by different binding affinities, for a total of 195 complexes. It stems from the need for a common benchmark to better assess the strength and weakness of scoring functions, decoupling them from the conformational search algorithm. Indeed, the benchmark provides already pre-calculated poses using different docking algorithms (namely GOLD [21], Surflex [22], and MOE [23]). The purpose of the present work was to use the CASF-2013 benchmark to evaluate whether the contact frequencies between fragments of ligands and protein residues could be used to identify the correct ligand binding regions and to distinguish between true and false ligands. To do this, a score, based on the contact frequencies of the ligand’s fragments, was assigned to each pose by following these steps: the pose was decomposed by BRICS and the fragment-residue contacts identified; then, each fragment was assigned a score based on the frequency in the training set of the contact identified. The final score for the pose was thus simply the sum of the scores of each fragment:(1)$$\overset{n}{\underset{i=1}{\sum }}f_{i}$$
where *f_i_* is the frequency of the contact identified for the fragment *i* in the dataset and *n* is the number of fragments of the ligand. The analysis of the results and the comparison with the results of other scoring functions (taken from ref. [11]) were performed using the script distributed together with the CASF-2013 benchmark dataset.

### 4.5. DUD-E

The Directory of Useful Decoys Enhanced (DUD-E) is a collection of protein targets, active compounds, and decoys. For each active compound, up to 50 decoys are presented. These decoys are selected in order to be similar from a physical point of view, but topologically different. For this work all the different protein family subsets were downloaded and after a cleaning procedure, which aimed to remove redundant structures and those which could not be treated by AutoDock Vina without a preliminary preparation (e.g., improper atom names, etc.), a final test set containing 87 structures was obtained. The structures were manually inspected, and in the case of pur2, a residual portion of a chain belonging to a second protein presented in the PDB file was removed. The docking simulations were carried out by using AutoDock Vina with default parameters, no residues were considered flexible, and the search space was extended to the entire protein. The output was rescored using Equation (1). The Root Mean Score Deviation (RMSD) values between the top scoring poses predicted by molecular docking (both by AutoDock Vina and the CS method reranking) were calculated through PyMol 1.8.1 [32] by using the command rmsd_cur. This method computes the RMSD between two atom sets without performing any spatial transformations. The ΔRMSD between the top scoring pose and the co-crystallized ligand is calculated through Equation (2):(2)$$\Delta \mathrm{RMSD}\;=RMSD_{CS}-RMSD_{ADV}$$
where RMSD_CS_ and RMSD_ADV_ are the RMSD values calculated for the scoring pose predicted by the CS method and AutoDock Vina, respectively.

Based on the ΔRMSD obtained in the previous test, two structures were randomly selected for virtual screening simulations: one in which CS performs better than AutoDock Vina and the other in which AutoDock Vina performs better than CS.

In these simulations, 20 active compounds and 400 decoy compounds were randomly selected for each target from the active compound and decoy compound lists provided by DUD-E. The library of compounds was compiled by generating the starting conformation for each ligand using ETKDG, a distance geometry and knowledge-based algorithm implemented in RDKIT [33]. To evaluate the success rate of the virtual screening test, the enrichment factor was calculated in the same way in which it is calculated in CASF:(3)$$EF=\frac{NTB_{n\%}}{NTB_{tot}\times n\%}$$
where EF is the enrichment factor, NTB_n%_ is the number of true binders observed in n% of the identified ligands, and NTB_tot_ is the total number of true binders for a given protein. N possible values in this test are 1, 5, and 10. 

### 4.6. AutoDock Vina

AutoDock Vina [12] is one of the most renowned and successful docking methods ever developed. It is derived from AutoDock 4 [34], which shares the same ligand/receptor preparation tools and input/output type, but achieves a greater speed and accuracy. AutoDock Vina’s scoring function consists of a weighted sum of steric interactions, hydrophobic contributions, and, where applicable, hydrogen bonding. All the docking simulations performed in this work are rigid, meaning that only the flexibility of the ligand is considered. 

## Figures and Tables

**Figure 1 ijms-20-02499-f001:**
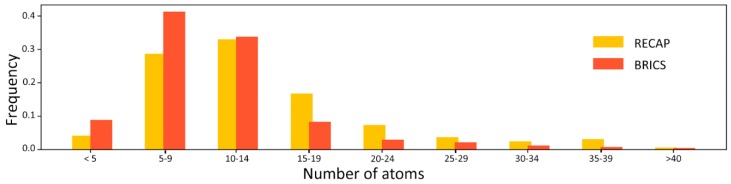
Distribution of the size of the fragments generated by RECAP (yellow bars) and BRICS (orange bars).

**Figure 2 ijms-20-02499-f002:**
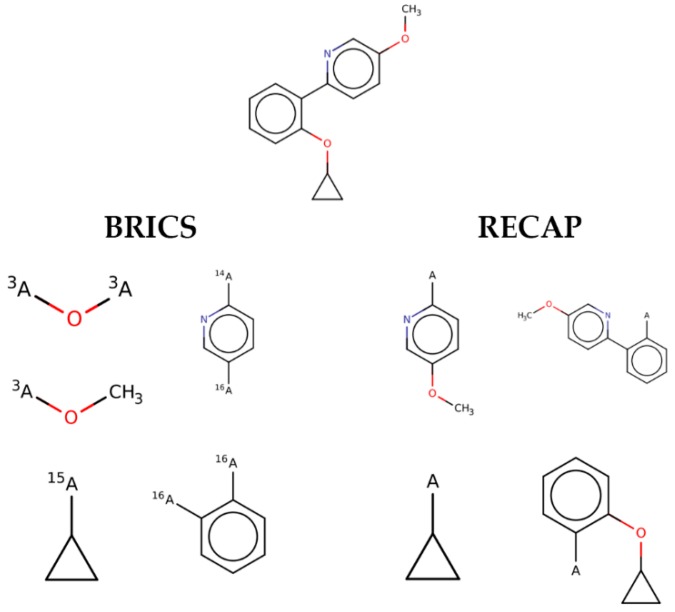
An example of the differences in the fragments generated by BRICS vs RECAPS. Note that, in contrast to RECAPS, the ends of the fragments generated by BRICS are marked with an isotopic label.

**Figure 3 ijms-20-02499-f003:**
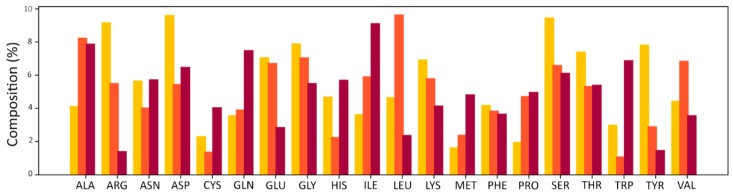
Amino acid composition of the binding pockets (yellow bars) with respect to that of the corresponding proteins (red bars) and that of all the known proteins in UniProt (orange bars).

**Figure 4 ijms-20-02499-f004:**
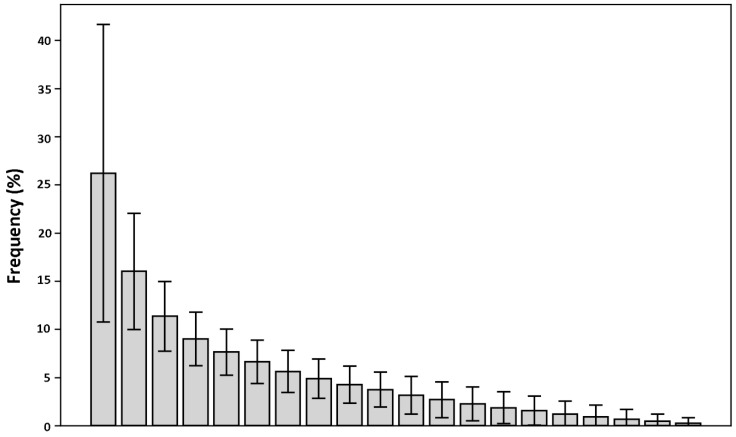
Distribution of the average residue-fragment contact frequencies. The error bars indicate the standard deviation (see text for details).

**Figure 5 ijms-20-02499-f005:**
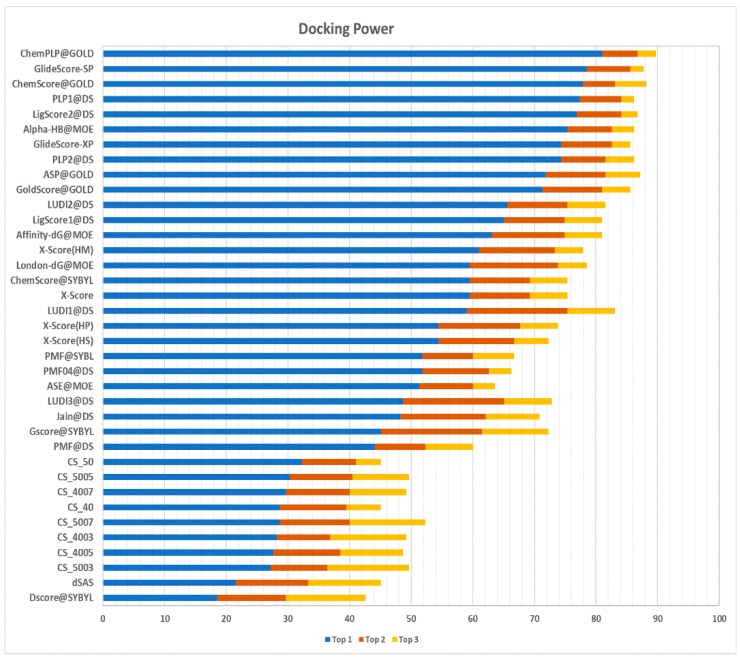
Comparison of the results of the CASF-2013 scoring functions and those based on the CS datasets in the docking power test ordered based on the presence of a native-like pose in the top one (blue bar), top two (orange bars), and top three (yellow bars) prediction(s).

**Figure 6 ijms-20-02499-f006:**
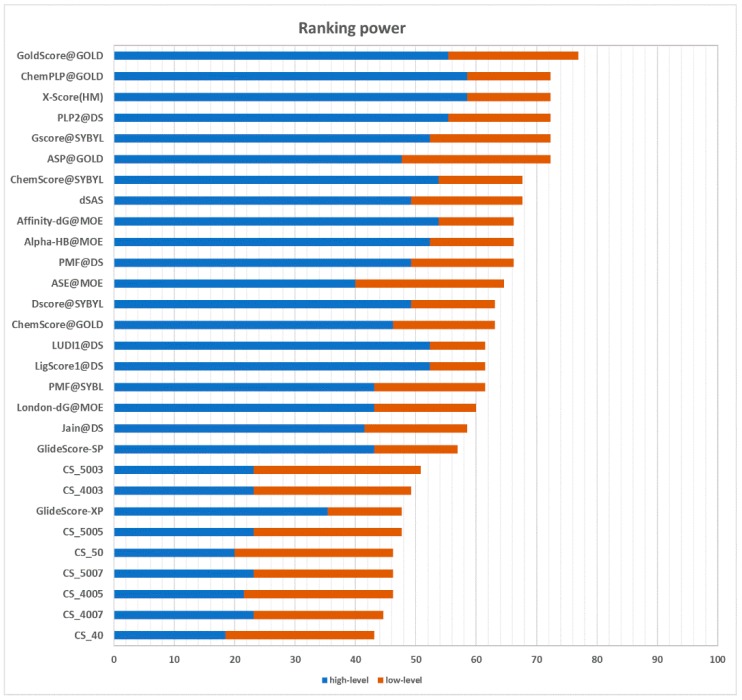
Success rates of scoring functions in the ranking power test for high-level prediction (blue bars) and low-level prediction (orange bars).

**Figure 7 ijms-20-02499-f007:**
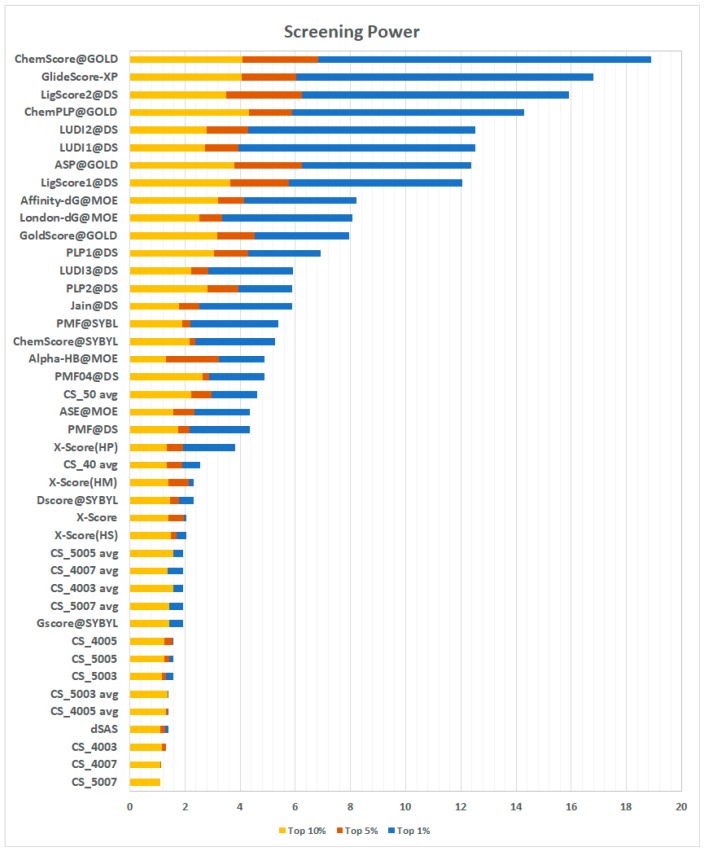
Results of the screening power test carried out on the CASF-2013 benchmark. In the picture, the results of the proposed method by employing both the sum and the average of the fragments’ contact frequencies (the latter identified by the suffix ‘avg’) are shown. The results are represented in terms of the enrichment factor achieved in the best scoring top 1% (blue bars), top 5% (orange bars), and top 10% (yellow bars) poses.

**Figure 8 ijms-20-02499-f008:**
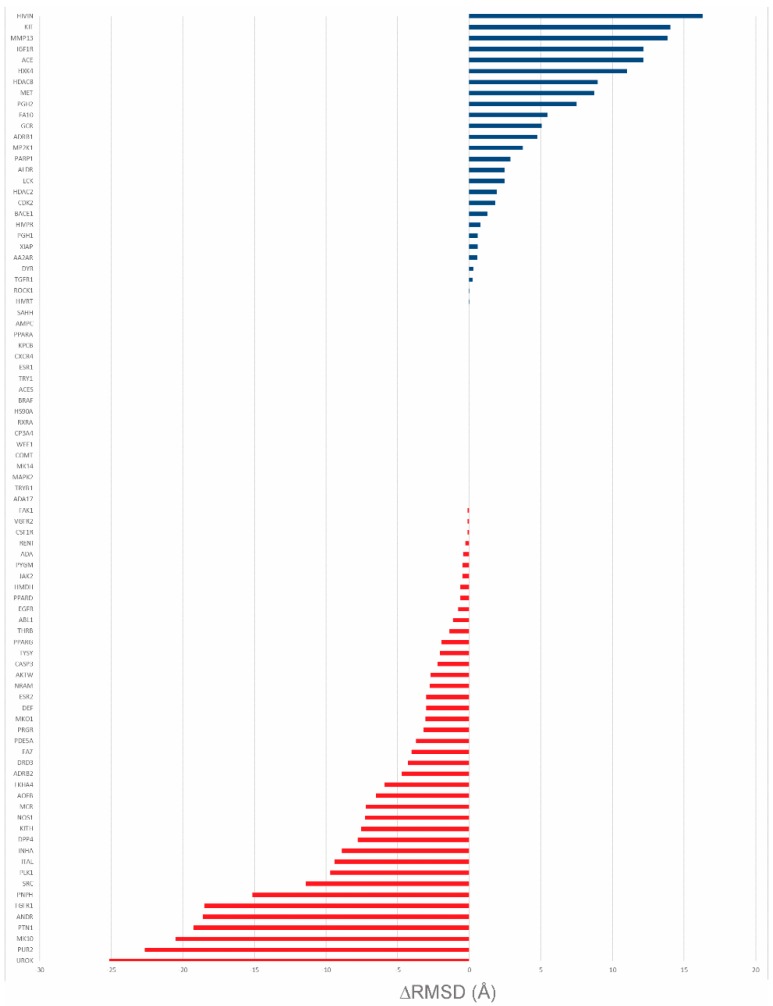
ΔRMSD comparison of AutoDock Vina and CS. Blue bars indicate cases in which the top scoring pose identified by CS is closer to the co-crystallized ligand than the one identified by AutoDock Vina. Red bars indicate the opposite cases.

**Figure 9 ijms-20-02499-f009:**
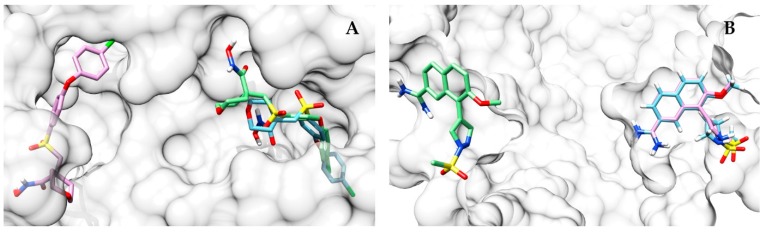
Comparison of the top scoring prediction of CS (green) and AutoDock Vina (pink) with respect to the co-crystallized ligand (light blue). In (**A**), the binding site of mmp13 (PDB: 830C) is depicted. It can be observed that CS predicts a near-native pose as top scoring, while AutoDock Vina places its top scoring pose far away from the binding pocket. The opposite scenario can be observed in panel (**B**) in the binding site of urok (PDB: 1SQT).

**Table 1 ijms-20-02499-t001:** Results of BRICS and RECAPS in the fragmentation of a non-redundant collection of ligands derived from the LIBRA database.

Fragmentation Method	Ligands	Fragments (unique)	Failures (%)
RECAP	2923	3978 (2107)	1486 (50.84%)
BRICS	2923	9427 (2539)	654 (22.37%)

**Table 2 ijms-20-02499-t002:** Datasets’ composition.

Dataset Set	Unique Proteins	Unique Ligands	Total Fragments	Total Complexes	Fragments>20
CS_5003	23955	13878	9212	26344	391
CS_5005	19471	10685	8044	21336	319
CS_5007	15373	7754	6322	16665	255
CS_4003	22990	13807	9194	25180	383
CS_4005	18450	10562	8012	20134	304
CS_4007	14298	7580	6251	15439	243
CS_50	7933	2923	2539	12003	141
CS_40	6579	2570	1734	9755	135

**Table 3 ijms-20-02499-t003:** Summary of the performance of the different CS datasets in the docking power test.

Data Set	Top1	Top2	Top3
%			
**CS_5003**	27.20	36.40	49.70
**CS_5005**	30.30	40.50	49.70
**CS_5007**	28.70	40.00	52.30
**CS_4003**	28.20	36.90	49.20
**CS_4005**	27.70	38.50	48.70
**CS_4007**	29.70	40.00	49.20
**CS_50**	32.30	41.00	45.10
**CS_40**	28.70	39.50	45.10

**Table 4 ijms-20-02499-t004:** Performance of the scoring functions in the scoring power test. Scoring functions are ranked based on the Pearson correlation coefficient.

Scoring Function	N	R^1^	SD
X-ScoreHM	195	0.614	1.78
ΔSAS	195	0.606	1.79
ChemScore@SYBYL	195	0.592	1.82
ChemPLP@GOLD	195	0.579	1.84
PLP1@DS	195	0.568	1.86
G-Score@SYBYL	195	0.558	1.87
ASP@GOLD	195	0.556	1.88
ASE@MOE	195	0.544	1.89
ChemScore@GOLD	189	0.536	1.90
D-Score@SYBYL	195	0.526	1.92
Alpha-HB@MOE	195	0.511	1.94
LUDI3@DS	195	0.487	1.97
GoldScore@GOLD	189	0.483	1.97
Affinity-dG@MOE	195	0.482	1.98
LigScore2@DS	190	0.456	2.02
GlideScore-SP	169	0.452	2.03
Jain@DS	191	0.408	2.05
PMF@DS	194	0.364	2.11
GlideScore-XP	164	0.277	2.18
CS_5003	146	0.252	2.26
CS_4003	146	0.249	2.26
London-dG@MOE	195	0.242	2.19
CS_5005	146	0.241	2.27
CS_4005	146	0.237	2.27
PMF@SYBYL	191	0.221	2.20
CS_50	146	0.162	2.30
CS_4007	146	0.145	2.31
CS_5007	146	0.140	2.31
CS_40	146	0.133	2.31

**Table 5 ijms-20-02499-t005:** Enrichment factor values obtained in the virtual screening tests carried out on mmp13 and urok using the different contact frequency datasets and AutoDock Vina.

	mmp13			urok		
	top1%	top5%	top10%	top1%	top5%	top10%
**CS_5003**	15.0	5.0	3.5	15.0	7.0	5.0
**CS_5005**	15.0	5.0	3.5	10.0	6.0	3.5
**CS_5007**	10.0	7.0	4.5	10.0	6.0	4.0
**CS_4003**	15.0	4.0	3.5	15.0	7.0	5.0
**CS_4005**	15.0	6.0	3.5	10.0	6.0	3.5
**CS_4007**	15.0	7.0	4.0	10.0	6.0	4.0
**CS_50**	0.0	3.0	2.0	5.0	3.0	2.0
**CS_40**	0.0	2.0	2.0	5.0	3.0	2.0
**ADV**	0.0	3.0	1.5	0.0	2.0	1.0

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
