# Peer review of "Fragment-Based Ligand-Protein Contact Statistics: Application to Docking Simulations"

_ijms, 2019, doi:10.3390/ijms20102499_

Round 1

Reviewer 1 Report

A brief summary

The manuscript described a novel work extensively verifying that there exists the preference of amino acid composition of binding pockets and that a simple contact frequency-based statistic potential may yield promising results comparable to those obtained with other more complex scoring functions. The extensive assessment of such simple potential functions is needed by the community and will benefit the wider community where people may utilize the results reported in the manuscript to do non-docking research instead of protein-compound docking research. However, as emphasized by authors in their manuscript, the core contribution of this work is just the proof of concept instead of proposing new powerful scoring function for protein-compound docking.

Broad comments

The authors did a good job to collect the benchmark data set excluding sequence-similar protein targets.

Specific comments

No.

Author Response

Rome, 16 may 2019

Prof. Dr. Giulio Vistoli

Guest Editor

Dear Editor,

please find enclosed the revised version of the manuscript entitled “Fragment-based ligand-protein contact statistics: application to docking simulations” by Gabriele Macari, Daniele Toti, Carlo del Moro and myself which we would like to submit to the International Journal of Molecular Sciences Special Issue “New Avenues in Molecular Docking for Drug Design”.

We have dealt with all the reviewers’ comments and a point to point reply can be found at the bottom of this letter.

We hope that the manuscript can be published in the International Journal of Molecular Sciences in its present form.

Best regards

                          Fabio Polticelli

Reply to the Reviewers’ comments

Reviewer 1

We thank very much the Reviewer for appreciating the spirit of our manuscript and for acknowledging that an extensive testing of our hypothesis has been carried out.

Reviewer 2

We thank very much the Reviewer for providing advice and comments that helped to improve the quality of our manuscript. Changes to the manuscript text are detailed below:

Reviewer’s comment

…The most important remark is that in my opinion the Authors should consider attaching the Python scripts used in the study as a supplementary information…

Authors’ reply

We agree with the Reviewer that making the Python scripts accessible, reusable and citable is in the interest of the scientific community and in ours too. However, since the scripts contain many lines of code and are based on custom modules, we have preferred to upload them on our lab website as a .zip file and provide a link for download:

that has been included in the Methods section of the revised manuscript’s text.

Reviewer’s comment

Line 26-29: "Computational approaches..." - this sentence sounds unclear, should be rewritten

Authors’ reply

The sentence has been rewritten as follows: ”The prediction and quantification of the strength of these interactions can be addressed by computational approaches, among which molecular docking simulations play a pivotal role [1]”

Reviewer’s comment

Line 52 - "exhaustive" is too much said, when we consider protein-ligand complexes. Perhaps Authors meant available, experimentally resolved protein-ligand complexes - in such case exhaustive study is possible. The sentence should be modified.

Authors’ reply

The sentence has been modified following the Reviewer’s advice: “…systematic study of a large collection of available, experimentally resolved protein-ligand complexes.”

Reviewer’s comment

Line 58/59 - "the these"

Authors’ reply

Thanks for pointing out this error, “the” has been removed from the text

Reviewer’s comment

Line 80 - In by pdb there is a sentence "Error! reference source not found" in this line.

Authors’ reply

We think this was due to a hypertext link that has now been removed. The manuscript text now appears correct also after conversion to .pdf

Reviewer’s comment

Line 323 - "Soga's work..." - the sentence should be corrected

Authors’ reply

The sentence was actually awkward and we thank the Reviewer for pointing out. The sentence has been corrected in the revised manuscript version:” In the work by Soga and coworkers, all the residues surrounding a ligand within 4.5 Å are considered part of the binding sites”

Reviewer’s comment

Line 324 - "fragments' closest residue" doesn't sound good, it should be rewritten in a more elegant way

Authors’ reply

Also this sentence has been rewritten as follows: “this work, instead, takes into account only the residue closest to the ligand fragment within 4.5 Å.”

Reviewer’s comment

Line 342 - experimentally-determined - the dash is not necessary

Authors’ reply

The dash has been removed

Reviewer’s comment

Line 428 - full-stop is missing

Authors’ reply

The full stop has been added

Reviewer’s comment

Line 434 - Modification is needed, e.g. "Complexes whose structures were"

Authors’ reply

The sentence has been modified following the Reviewer’s advice:"Complexes whose structures were determined by NMR were not included in the dataset.”

Reviewer’s comment

Line 487 - should it be: "...frequency of the contact identified for the fragment i in the dataset"?

Authors’ reply

Sure, thanks for suggesting a clearer sentence. The text has been modified accordingly.

Reviewer’s comment

Line 512 - "two structures"

Authors’ reply

The sentence was actually unclear and, indeed, unnecessary as it did not provide any useful detail to the reader. Thus it has been removed from the revised manuscript’s text.

Reviewer 2 Report

The manuscript of Macari et al. presents an investigation of patterns of interactions between proteins and their ligands. The concept is interesting, as it shows that despite the variety of ligand structures and protein sequences, the protein-ligand complexes usually follow some simple, universal patterns of side chain-functional group interactions. The study is properly designed, the outcomes can be considered inspiring, and the manuscript doesn't have any serious flaws. I have only some comments regarding grammar, punctuation and style. The most important remark is that in my opinion the Authors should consider attaching the Python scripts used in the study as a supplementary information. Although not necessary, sharing them with community could contribute to development of other applications of the concept, accelerate research in the field and therefore increase the number of citations.

Minor comments:

Line 26-29: "Computational approaches..." - this sentence sounds unclear, should be rewritten

Line 52 - "exhaustive" is too much said, when we consider protein-ligand complexes. Perhaps Authors meant available, experimentally resolved protein-ligand complexes - in such case exhaustive study is possible. The sentence should be modified.

Line 58/59 - "the these"

Line 80 - In by pdb there is a sentence "Error! reference source not found" in this line.

Line 323 - "Soga's work..." - the sentence should be corrected

Line 324 - "fragments' closest residue" doesn't sound good, it should be rewritten in a more elegant way

Line 342 - experimentally-determined - the dash is not necessary

Line 428 - full-stop is missing

Line 434 - Modification is needed, e.g. "Complexes whose structures were"

Line 487 - should it be: "...frequency of the contact identified for the fragment i in the dataset"?

Line 512 - "two structures"

Author Response

(The authors gave the same response as above.)
